# All-Dielectric Gratings with High-Quality Structural Colors

**DOI:** 10.3390/nano13172414

**Published:** 2023-08-25

**Authors:** Tongtong Wang, Yuanhang Zhao, Bo Yu, Mingze Qin, Zhihui Wei, Qiang Li, Haolong Tang, Haigui Yang, Zhenfeng Shen, Xiaoyi Wang, Jinsong Gao

**Affiliations:** 1Key Laboratory of Optical System Advanced Manufacturing Technology, Changchun Institute of Optics, Fine Mechanics and Physics, Chinese Academy of Sciences, Changchun 130033, China; wangtt@ciomp.ac.cn (T.W.); weikmp@163.com (Z.W.); liqiang@ciomp.ac.cn (Q.L.); tanghl@ciomp.ac.cn (H.T.); yanghg@ciomp.ac.cn (H.Y.); zf_shen@163.com (Z.S.); wangxiaoyi1977@sina.com (X.W.); gaojs999@163.com (J.G.); 2College of Da Heng, University of the Chinese Academy of Sciences, Beijing 100039, China; 3State Key Laboratory of Applied Optics, Changchun Institute of Optics, Fine Mechanics and Physics, Chinese Academy of Sciences, Changchun 130033, China; 4Jilight Semiconductor Technology Co., Ltd., Changchun 130033, China; qinmingze1219@163.com

**Keywords:** all-dielectric gratings, magnetic dipole resonance, CIE 1931 chromaticity diagram, structural color

## Abstract

We present a dual-layer hafnium dioxide (HfO_2_) grating capable of full-color modulation in the visible spectrum by leveraging the magnetic dipole resonance induced by the lower-layer HfO_2_ grating, while the upper-layer HfO_2_ grating serves as a refractive index matching layer to effectively suppress high-order Mie resonances at shorter wavelengths. The HfO_2_/HfO_2_ grating exhibits a significantly larger distribution area in the CIE 1931 chromaticity diagram compared to the HfO_2_ grating. Furthermore, the structural color saturation closely approximates that of monochromatic light. Under varying background refractive index environments, this structure consistently exhibits high-quality structural color. However, the hue of the structural color undergoes alterations. When the polarization angle is below 20°, the saturation of the acquired structural color remains remarkably consistent. However, exceeding 20° results in a significant degradation in the quality of the structural color. This study demonstrates the promising potential for diverse applications, encompassing fields such as imaging and displays.

## 1. Introduction

The wavelength range from 400 nm to 700 nm is referred to as the visible light range, as electromagnetic waves within this range can stimulate photoreceptor cells on the human retina, leading to the perception of colors [1,2,3,4]. Traditional pigments have been extensively utilized in object decoration and color printing over the past three decades. However, these pigments possess several drawbacks, including environmental pollution [5], potential carcinogenic effects [6], and limitations in image-sensing applications due to low resolution and inadequate color quality [7]. With continuous research and expansion in the metasurface field, the investigation of structural color based on metasurfaces has gained significant attention. The advancement of micro–nanostructure fabrication techniques propels progress in this field. For instance, the utilization of ultraviolet/visible light for photopatterning has enabled the fabrication of photonic crystals with varied dimensions and sizes [8,9,10,11,12,13,14]. Structural color arises from the scattering, reflection, diffraction, and interference of light in the micro–nanostructures, without the involvement of specific pigments [15]. It possesses various advantages, including high resolution, superior contrast, exceptional color stability, minimal power consumption, and recyclability, thereby exhibiting the potential to entirely supplant conventional pigment coloring [16,17,18,19]. Such structural colors can also be found in nature, such as in the wings of butterflies, the feathers of peacocks, and the shells of beetles. Plasmonic metasurfaces and dielectric metasurfaces can both achieve full-color modulation within the visible light range [20,21,22,23,24,25,26,27,28,29], but the light absorption caused by plasmonic metasurfaces during the light modulation process significantly diminishes the saturation of structural colors. Consequently, several researchers have advocated the utilization of high refractive index materials (Si, TiO_2_, and Si_3_N_4_) to attain robust dipole resonance, thereby substituting plasmonic metasurfaces characterized by high losses. With the rapid development of nanofabrication technology, researchers have explored metasurface structures with excellent structural color results, such as gratings [30], nanoblocks [31], and nanodisks [32].

Hue, brightness, and saturation are utilized to assess color quality [33,34]. The quality of structural colors is influenced by several factors, including the peak position, spectral bandwidth, suppression of high-order Mie resonances, and peak intensity. These parameters play distinct roles in defining different aspects of the structural color appearance. Specifically, the peak position contributes to the hue of the structural color, while the bandwidth and high-order Mie resonances are critical for constraining the saturation. Additionally, the peak intensity directly determines the brightness of the structural color. Although the excitation of a strong dipole resonance by high refractive index materials can generate structural colors encompassing the full range of hues, the presence of high-order Mie resonances within a limited wavelength range hampers the enhancement of color saturation, particularly in the production of red. Currently, the methods to solve this problem include the utilization of Rayleigh scattering for the mitigation of high-order Mie resonance modes in the short wavelength range [35], and the incorporation of a matching layer on either the upper or lower surface of the Mie scatterer to suppress the multipole modes at non-resonant wavelengths [36]. Thus, presenting a concise and efficacious approach to enhance color saturation through the synergistic integration of prevailing techniques stands as a pivotal quandary to be addressed.

Combining the narrow bandwidth, high efficiency, and low side-peak interference spectral characteristics of grating structures, we investigate the suppression of high-order Mie resonance in the double-layer HfO_2_/HfO_2_ grating structure. Unlike previous research reports, the refractive index matching layer and the material of the Mie scatterer were completely identical. Comparing the chromaticity coordinates of single-layer HfO_2_ gratings with those of the double-layer HfO_2_/HfO_2_ gratings on the CIE 1931 chromaticity diagram confirms the effectiveness of this method for enhancing color saturation. This finding holds significant implications for high-precision imaging, advanced camera technologies, and color printing applications.

## 2. Model Building and Analysis

In the selection of materials, we chose hafnium dioxide (HfO_2_) as the material for the grating structure. HfO_2_ is an exceptional optical material known for its low stress, high hardness, high refractive index, and negligible extinction coefficient. Moreover, HfO_2_ nanostructures have demonstrated the ability to exhibit resonances associated with electric and magnetic dipoles during Mie scattering. The double-layer of HfO_2_ gratings was fabricated on a glass substrate, with the period (P), size (D_1_, D_2_), and thickness (H_1_, H_2_) of the structural units, as depicted in Figure 1. The upper layer of the HfO_2_ gratings functions as a refractive index matching layer. Both the upper layer and the air in the grating gaps are considered an ‘artificial material’ with an equivalent refractive index of *n_eff_*, Calculation of the effective refractive index is based on the two-dimensional Bruggeman equation:(1)$$F\left(\frac{n_{eff}-n_{1}}{n_{eff}+n_{1}}\right)+\left(1-F\right)\left(\frac{n_{eff}-n_{0}}{n_{eff}+n_{0}}\right)=0$$
where, *n*_1_ represents the refractive index of the material, *n*_0_ denotes the refractive index of air, and *F* corresponds to the fill factor. Thus, when the refractive index of the matching layer is the same as that of the Mie scattering material, it also has the effect of suppressing high-order Mie resonances.

To accurately analyze the intensity, position, and electromagnetic field distribution of the reflection peak resulting from normal incidence of a plane wave onto the structural system, we conducted FDTD simulations to simulate the grating structure. The incoming light is polarized in the transverse magnetic (TM) mode, with its polarization direction perpendicular to the -Z axis within the system. Owing to the height symmetry exhibited by the structural components along the X and Y axes, periodic boundary conditions were enforced in both of these directions. In order to eliminate echo interference, a perfectly matched layer (PML) was introduced in the Z direction simultaneously. A high mesh accuracy (mesh accuracy = 6) was employed to ensure the convergence of the simulation results.

The upper-layer HfO_2_ grating’s structural parameters inevitably impact the final response of the reflection spectrum, serving as a refractive index matching layer. First, we investigated the effect of the size of the refractive index matching layer on the suppression of high-order Mie resonances. The lower HfO_2_ grating is fixed at a period P of 385 nm, thickness H_1_ of 140 nm, and size D_1_ of 220 nm, while the upper HfO_2_ grating is fixed at a period P of 385 nm and thickness H_2_ of 100 nm. Figure 2a depicts the correlation between the spectral response and the upper grating size D_2_, with D_2_ ranging from 0 to 220 nm. Increasing the size D_2_ of the upper grating beyond 100 nm results in a redshift of the reflection peak and a significant enhancement in peak intensity, implying that the dimensions of the upper grating have a substantial impact on the brightness and hue of the structural color. Figure 2b presents the reflection spectra of the system under varying sizes, specifically D_2_ values of 0 nm, 120 nm, and 220 nm. Compared with the other two sizes, for a size D_2_ of the upper grating at 120 nm, the reflection spectrum exhibits characteristics of a narrow bandwidth, high efficiency, and low side-peak interference. Choosing appropriate structural parameters can effectively suppress higher-order Mie resonance modes, leading to enhanced saturation of the structural color.

Next, we investigated the influence of the thickness of the refractive index matching layer on the high-order Mie resonances. The structure parameters of the lower HfO_2_ grating were identical to those depicted in Figure 2, with the upper HfO_2_ grating size D_2_ fixed at 120 nm. As the thickness of the upper grating increases, the position of the reflection peak remains unchanged, while the bandwidth of the reflection spectrum gradually widens and eventually stabilizes at a specific value, as illustrated in Figure 3a. This indicates that the thickness of the upper grating affects the brightness of the structural color but has no effect on the hue. By comparing the reflection spectra of the system with and without the upper HfO_2_ grating, the transmittance of the non-resonant peak positions significantly increases and effectively suppresses the impact of high-order Mie resonances in the short wavelength range when an appropriate thickness of the refractive index matching layer is selected. At the same time, the peak intensity increases from 70% to 99.9%, achieving nearly perfect reflection. Optimizing the thickness and size of the upper HfO_2_ grating can enhance the brightness and saturation of the structural color.

Finally, we investigated the spectral modulation characteristics of the structural system across the full spectrum and obtained the optimal structural parameters through a comprehensive series of data simulation experiments. The upper HfO_2_ grating was fixed at a thickness of 100 nm, while the lower HfO_2_ grating was fixed at a thickness of 140 nm. The ratio of the sizes of the upper and lower HfO_2_ gratings was 44/23 (D_1_/D_2_), and the ratio of the size to the period of the lower grating was 4/7 (D_2_/P). In order to compare the overall spectral response of the system with and without an upper HfO_2_ grating, different periods were scanned, as shown in Figure 4a,c. Regardless of the existence of a refractive index matching layer, the structural system demonstrates complete spectral modulation within the operational bandwidth, thus enabling the realization of structural colors with a full range of hues. The reflectance spectra of the system at periods of 266 nm, 308 nm, 350 nm, 392 nm, and 434 nm are depicted in Figure 4b,d. The influence of incorporating an upper HfO_2_ grating to mitigate high-order Mie resonances remains inconspicuous within the reflectance spectra of the near-working band short-wave range, as high-order Mie resonances are absent in the working band. Consequently, the reflectance spectra with and without the HfO_2_ grating exhibit near-identical profiles. As the reflection peak approaches longer wavelengths, the suppression of high-order Mie resonances becomes increasingly apparent, and the reflection peak maintains a near-perfect intensity of reflection. Compared with the HfO_2_ grating, the HfO_2_/HfO_2_ grating achieves high reflectivity and low side-peak interference while achieving full spectral modulation, indicating a significant improvement in the quality of structural colors with different hues.

To elucidate the physical mechanism underlying the reflection peak within the operational band of HfO_2_ and HfO_2_/HfO_2_ gratings, we scrutinized the magnetic field and the electric field distribution at the resonant peaks with identical periodicity. When the period P is 280 nm, the peak wavelengths of the two grating structures are 446 nm. Figure 5a depicts the magnetic field energy distribution in the XOZ plane for HfO_2_ gratings. The magnetic field energy is predominantly localized within the lower HfO_2_ grating. In Figure 5b, the depiction presents the spatial distribution of the electric field energy confined within HfO_2_. When the incident plane wave traverses the HfO_2_ grating structure, the electric field energy becomes localized around the regions of high refractive index material. This localization displays a pronounced orthogonal correlation with the distribution of the magnetic field. In the lower HfO_2_ grating, the electric field vector exhibits a circular distribution, which induces magnetic dipole resonance through circular displacement currents generated by incident electromagnetic waves, as shown in Figure 5c. Figure 5d portrays the magnetic field distribution at the resonance peak of the HfO_2_/HfO_2_ grating. Analogous to the monolayer HfO_2_ grating, the magnetic field energy is effectively confined within the lower grating layer. This strong resonance mode occupies an absolute position in the spectral response, and its advantages can be effectively utilized in the field of optical device modulation. Although the dominant magnetic dipole resonance governs the reflection spectrum, the high-order Mie resonance mode exerts a more substantial impact on the quality of structural color at shorter wavelengths. This high-order mode does not actually appear within the working band range, and its resonance peak is below 400 nm (less than the working band), but its peak position exhibits a red shift with increasing period.

## 3. Results and Discussion

The CIE 1931 color space is the first mathematical method used to describe color. Colors were assessed by calculating their chromaticity coordinates and plotting them in the chromaticity space. The quality of colors was evaluated based on the position and distribution area of the chromaticity coordinates within the chromaticity space. The numerical values of the three stimuli (*X*, *Y*, *Z*) are defined and obtained by the CIE tristimulus formula [29]:(2)$$X=K\int S(\lambda )\bar{x}(\lambda )R(\lambda )d\lambda$$
(3)$$Y=K\int S(\lambda )\bar{y}(\lambda )R(\lambda )d\lambda$$
(4)$$Z=K\int S(\lambda )\bar{z}(\lambda )R(\lambda )d\lambda$$
(5)$$K=100/\int S(\lambda )\bar{y}(\lambda )d\lambda$$
where, *R*(λ) represents the spectral distribution of reflected light from a surface after being illuminated with a light source, *S*(λ) represents the energy distribution of the illuminating light source, $\bar{x}(\lambda )$, $\bar{y}(\lambda )$, $\bar{z}(\lambda )$ are the color-matching functions of the CIE standard observer under a 2° visual field, and *d*λ is the integration interval. The chromaticity coordinates in the two-dimensional plane were determined by calculating the three-stimulus values:(6)$$x=\frac{X}{X+Y+Z}$$
(7)$$y=\frac{Y}{X+Y+Z}$$

The chromaticity coordinates of the structural colors obtained from two different structural systems were calculated for periods of P = 266 nm, 308 nm, 350 nm, 392 nm, and 434 nm. The resulting coordinates were then plotted on the CIE 1931 chromaticity diagram, as shown in Figure 6. “O” represents the chromaticity coordinates of the HfO_2_ grating structure, and “■” represents the chromaticity coordinates of the HfO_2_/HfO_2_ grating structure. The edge of the chromaticity diagram is formed with monochromatic light, and the closer the chromaticity coordinates are to the edge, the higher the purity and saturation of that color. The blue chromaticity coordinates acquired from both structures exhibit remarkable similarity, whereas substantial distinctions emerge in the red and green chromaticity coordinates. The suppression of the high-order Mie resonant modes by the upper HfO_2_ grating causes the red and green colors to approach the edge of the chromaticity diagram. The color gamut area obtained from the HfO_2_/HfO_2_ grating structure is much larger than that of the HfO_2_ grating alone. It determines the number of colors, which is critical for LCD displays to render a wider spectrum of colors.

Considering that the refractive index of the background in practical applications may differ, which can influence the spectral response, the actual structural color may exhibit a distinct hue compared to the simulated structural color. To address this concern, we investigated the background refractive index by keeping the other structural parameters fixed while varying the refractive index (n) of the environment. The range of n was set from 1 to 1.5, as shown in Figure 7a. The reflection peak exhibited a red shift phenomenon as the background refractive index was incrementally increased, with a maximum peak shift of 49 nm. Figure 7b shows the relationship between the reflection and peak half-width at different resonant peak positions of the background refractive index at 1.0, 1.1, 1.2, 1.3, and 1.4. The peak intensity remained unaffected by the background refractive index, exhibiting nearly 100% reflectivity. The peak half-width exhibited the narrowest measurement at the peak background refractive index, measuring approximately 3.5 nm. To more clearly elucidate the influence of the background refractive index on the structural color, we compared the reflection spectra of the structural system in air (n = 1.0) and water (n = 1.33), as shown in Figure 7c. In both environments, the reflection spectra exhibited narrowband characteristics, high efficiency, and minimal interference from side peaks. The separation between the two peak positions measured approximately 25 nm, resulting in a shift in the chromaticity coordinates of the structural color on the CIE 1931 chromaticity diagram, as shown in Figure 7d. This finding highlights the sensitivity of the structural system’s spectral response to environmental changes, emphasizing the importance of considering and adjusting environmental factors when achieving a desired structural color.

To ascertain the efficacy of upper HfO_2_ gratings in mitigating high-order Mie resonances across diverse background environments, we performed a comparative analysis of the structural color fidelity achieved with the HfO_2_ gratings and HfO_2_/HfO_2_ gratings immersed in water, aiming to evaluate their respective performance. The structural parameters of both gratings were set to match those shown in Figure 4. The reflection spectra were plotted for periods of P = 259 nm, 280 nm, 301 nm, 322 nm, 343 nm, 364 nm, 385 nm, 406 nm, and 427 nm, as shown in Figure 8a,b. Compared to the reflection spectra of the HfO_2_ gratings in air, the spectra in water demonstrate a remarkable enhancement in both peak intensity and bandwidth, attributed to the utilization of water as a refractive index matching layer. Although the bandwidth of the reflection spectra of the HfO_2_ gratings is evidently narrower compared to that of the HfO_2_/HfO_2_ gratings, the color coordinates of the latter align more closely with the edge contour in the chromaticity diagram. The color gamut enclosed by the HfO_2_/HfO_2_ gratings’ RGB primaries exhibits a significantly larger range than that of the HfO_2_ gratings, as shown in Figure 8c. For blue, the color coordinates of both structures exhibit remarkable similarity. For red and green, the structural colors obtained from the HfO_2_/HfO_2_ gratings exhibit a tendency towards monochromatic light.

The influence of the incident polarization angle on color quality can be employed as a method for color adjustment. Figure 9a illustrates the spectral reflectance for incident polarization angles ranging from 0° to 90°. As the polarization angle increases, we observe a slight shift in the peak position accompanied by a significant decrease in peak intensity. Simultaneously, the reflectance at non-resonant peak positions exhibits a gradual increase. We investigated the effect of various polarization angles on the quality of the structural color and plotted the resultant structural colors on the CIE 1931 chromaticity diagram, as shown in Figure 9b. When the polarization angle is below 20°, the chromaticity coordinates exhibit limited variation and approach the periphery of the chromaticity diagram, implying that the HfO_2_/HfO_2_ grating retains a remarkable saturation for the structural color achieved. When the polarization angle exceeds 20°, the structural color shifts towards green and the chromaticity coordinates progressively deviate from the contour edge. This observation indicates that an increased incident polarization angle results in significant degradation of the structural color quality. To further determine this degradation behavior, we compared the structural colors obtained at polarization angles of 0°, 30°, 60°, and 90° for different periods, as shown in Figure 9c. It can be observed that an increment in the polarization angle exerts a pronounced influence on the red hue, whereas the blue and green hues exhibit remarkable consistency at polarization angles below 30°.

## 4. Conclusions

In summary, we propose a HfO_2_/HfO_2_ grating structure that effectively suppresses high-order Mie resonances at shorter wavelengths, leading to enhanced color saturation in structural effects. The influence of the structural parameters of the top HfO_2_ grating on the spectral response is investigated, and the physical mechanism of the reflection peak in the working range is elucidated using the magnetic dipole resonance mode. Full-color modulation in the visible wavelength range is achieved by adjusting the grating period. The structure exhibits high reflectivity (~1), narrow bandwidth, and minimal sideband interference across varying refractive index backgrounds. In the CIE 1931 chromaticity diagram, the chromaticity coordinates of the HfO_2_/HfO_2_ grating are located proximal to the contour edge and encompass a substantial portion of the diagram’s area. When the incident light’s polarization angle is less than 20°, the system retains a high-quality structural color. This research has significant implications for applications in advanced imaging devices, optical security, and optical data storage.

## Figures and Tables

**Figure 1 nanomaterials-13-02414-f001:**
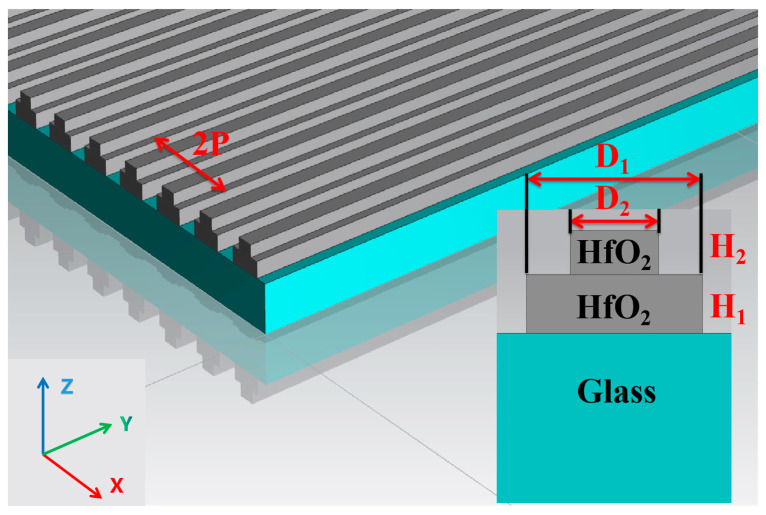
The schematic diagram of the structure of a double-layer HfO_2_/HfO_2_ grating.

**Figure 2 nanomaterials-13-02414-f002:**
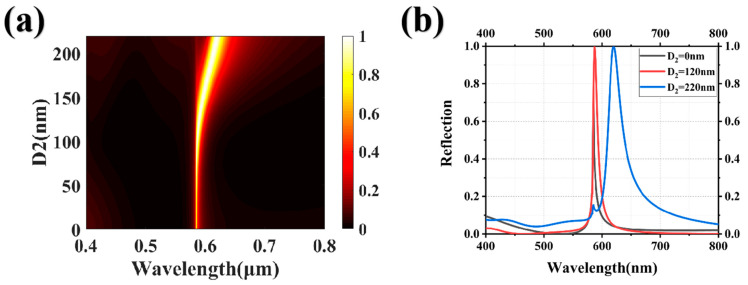
(**a**) The parameter sweep plot illustrating the upper HfO_2_ grating size D_2_, with a scanning range from 0 to 220 nm; and (**b**) the reflection curves corresponding to three distinct grating sizes, specifically D_2_ values of 0 nm, 120 nm, and 220 nm. The lower HfO_2_ grating is fixed at a period P of 385 nm, thickness H_1_ of 140 nm, and size D_1_ of 220 nm.

**Figure 3 nanomaterials-13-02414-f003:**
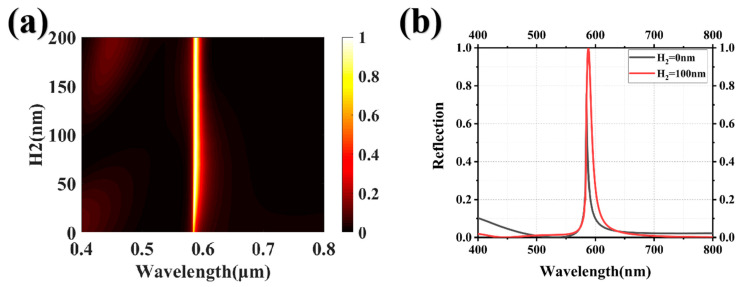
(**a**) The parameter sweep chart illustrating the upper HfO_2_ grating thickness H_2_, ranging from 0 nm to 200 nm; (**b**) the reflection curves of the structural system are shown for the grating thickness H_2_ at 0 nm and 100 nm.

**Figure 4 nanomaterials-13-02414-f004:**
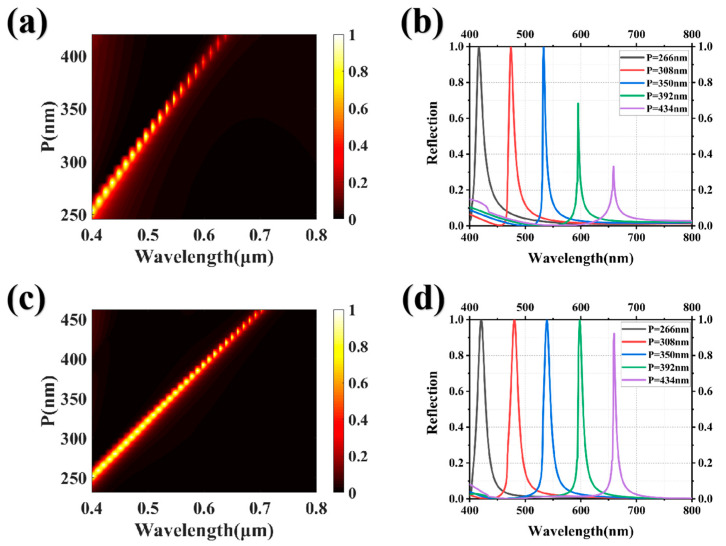
(**a**) The scanning diagram showcasing HfO_2_ gratings with varying periods; (**b**) the reflection spectra of HfO_2_ gratings with periods of 266 nm, 308 nm, 350 nm, 392 nm, and 434 nm are presented; (**c**) and (**d**) are the scanning diagram and reflection spectra, respectively, of the HfO_2_/HfO_2_ gratings with distinct periods.

**Figure 5 nanomaterials-13-02414-f005:**
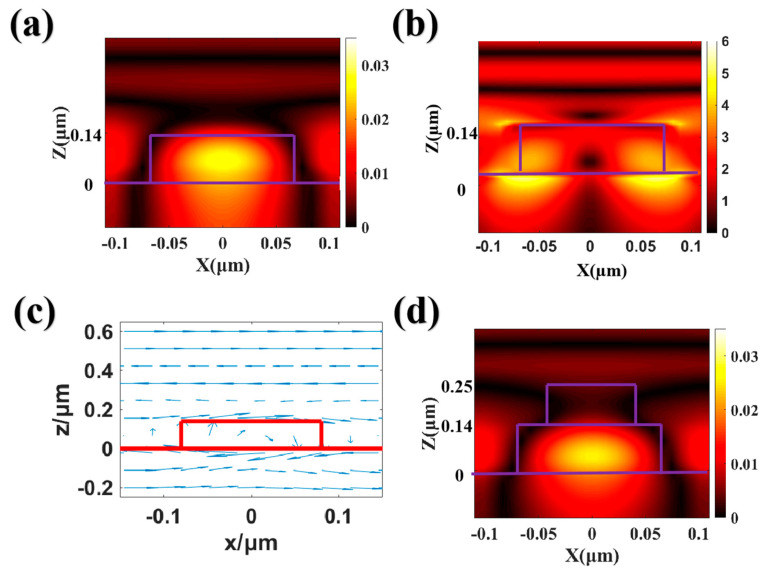
(**a**) the magnetic field and (**b**) the electric field energy distribution at the resonance peak position in the HfO_2_ grating, (**c**) the electric displacement vector distribution in the lower HfO_2_ grating, and (**d**) the magnetic field energy distribution at the resonance peak position in the HfO_2_/HfO_2_ grating.

**Figure 6 nanomaterials-13-02414-f006:**
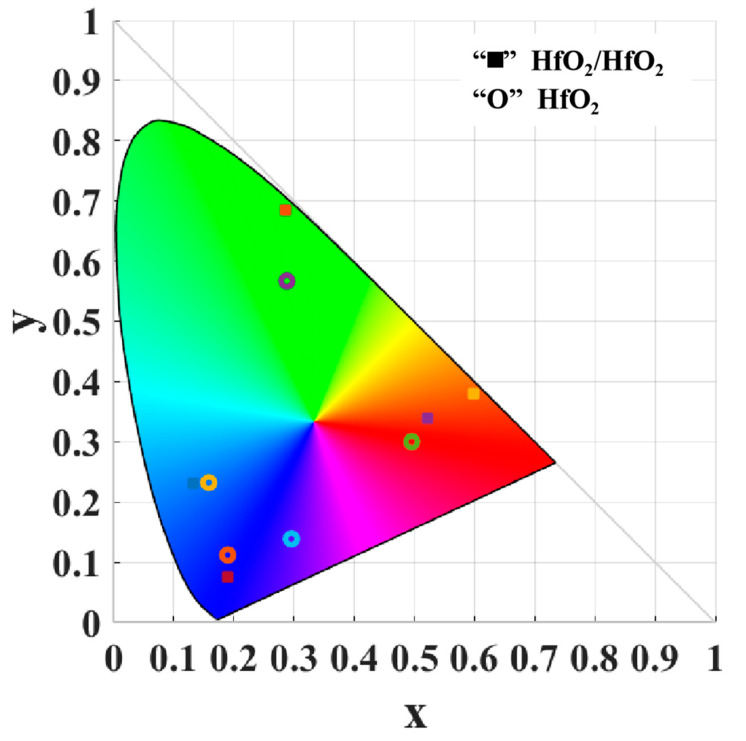
The chromaticity coordinates obtained from the HfO_2_/HfO_2_ and the HfO_2_ gratings are distributed in the CIE 1931 chromaticity diagram.

**Figure 7 nanomaterials-13-02414-f007:**
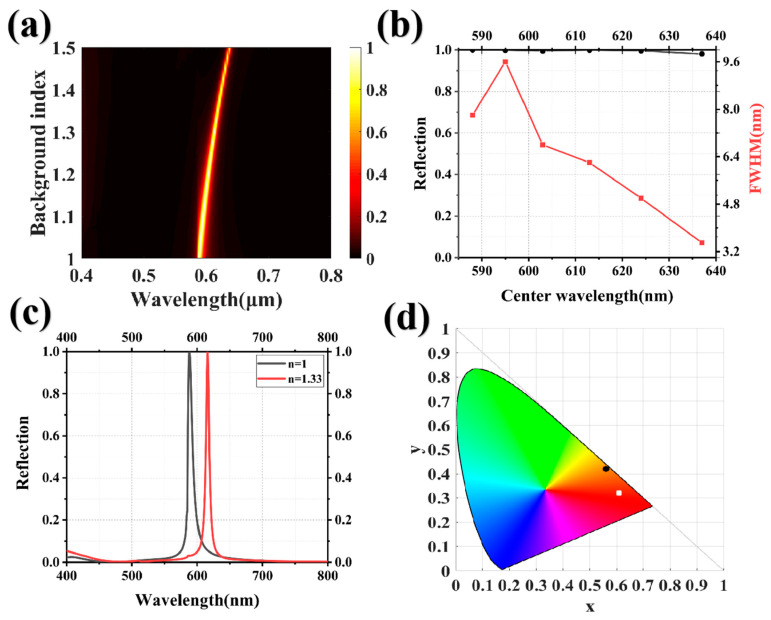
(**a**) the sweep chart of the background refractive index in the range of 1 to 1.5; (**b**) the relationship between the reflection rate and FWHM of the resonance peak at different background refractive indices is represented by the “black line” and “red line”, respectively; (**c**) the reflection spectra of HfO_2_/HfO_2_ grating in air and water; and (**d**) the distribution of chromaticity coordinates in the CIE 1931 chromaticity diagram, where the black dots represent the structural color in air, and the white dots represent the structural color in water.

**Figure 8 nanomaterials-13-02414-f008:**
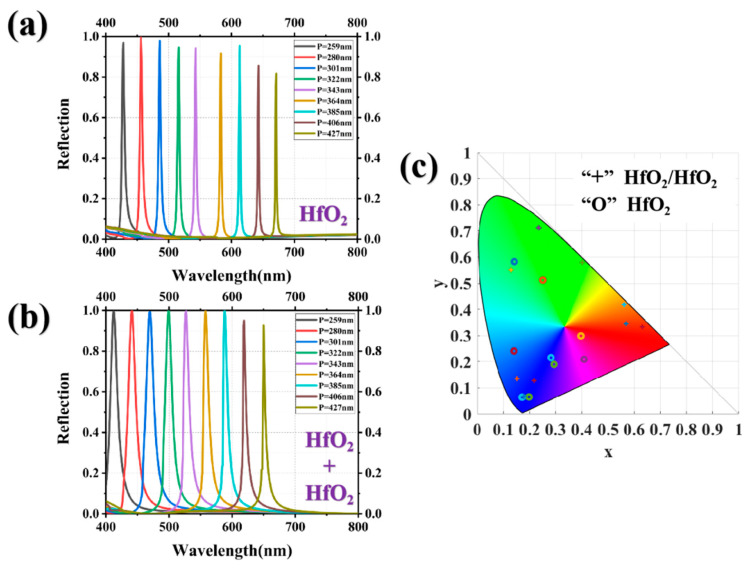
The reflection spectra of (**a**) the HfO_2_ grating and (**b**) the HfO_2_/HfO_2_ grating with different periods in water, (**c**) the chromaticity coordinates of the obtained structural colors of the two gratings in the CIE 1931 chromaticity diagram.

**Figure 9 nanomaterials-13-02414-f009:**
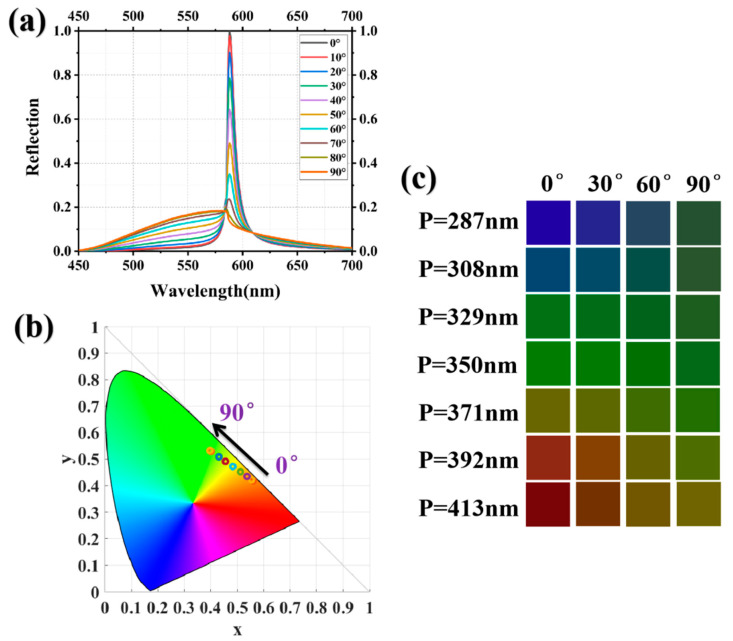
(**a**) the reflectance spectra of the HfO_2/_HfO_2_ grating at different polarization angles ranging from 0° to 90° with a sampling interval of 10°; (**b**) the chromaticity coordinates of the structural colors obtained at different polarization angles are plotted on the CIE 1931 chromaticity diagram; and (**c**) the structural colors obtained when the incident light is polarized at 0°, 30°, 60°, and 90° are compared for gratings with different periods.

## Data Availability

The data that support the plots within this paper and other findings of this study are available from the corresponding authors on reasonable request.

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
