# Peer review of "All-Dielectric Gratings with High-Quality Structural Colors"

_nanomaterials, 2023, doi:10.3390/nano13172414_

Round 1

Reviewer 1 Report

This paper is well-written and demonstrates a well-executed study. It not only presents a comprehensive investigation into structural color modulation using the dual-layer HfO2 grating structure, but also highlights the transformative potential of this innovative approach. The study is grounded in the utilization of the CIE 1931 color space and the CIE tristimulus formula, which serve as foundational mathematical frameworks for describing and evaluating structural colors. The paper seamlessly integrates its sections, progressing fluidly from setting the context to modeling exploration, results and discussion, and coherent conclusions. The incorporation of theoretical concepts, simulation outcomes, and practical applications enriches the reader's understanding of the proposed technology. The potential applications of this study in advanced imaging devices, optical security, and optical data storage position it as a noteworthy contribution to the field.

VERY MINOR REVISIONS:

a)      Before mentioning the acronym (HfO2), spell out the entire word (even in the abstract)

b)     Introduction. Remove the square bracket on line 5 of page 2: "...by high losses."

c)      In Figure 2, the caption should provide a reminder of the dimensions of D1, the period of the grating, and the thickness of D1.

d)     Additionally, the authors could include in the introduction a brief discussion on the approach involving photonic crystals of varying dimensionalities (1D gratings, 2D, and 3D structures) achieved through UV/Vis patterning of diverse materials, including polymers, and different thicknesses for many different applications (e.g., see:

1)     John, S. (1987) Strong localization of photons in certain disordered dielectric superlattices. Physical Review Letters, 58(23), 2486-2489;

2)     Yablonovitch, E. (1987) Inhibited spontaneous emission in solid-state physics and electronics. Physical Review Letters, 58(20), 2059-2062;

3)     Lucchetta D.E., Vita, F., Castagna, R., Francescangeli, O., Simoni, F. (2012) Laser emission based on first-order reflection by novel composite polymeric gratings. Photonics and Nanostructures - Fundamentals and Applications, 10(1), 140-145;

4)     Johnson, S.G., Joannopoulos, J.D. (2000) Three-dimensionally periodic dielectric layered structure with omnidirectional photonic band gap. Applied Physics Letters, 77(22), 3490-3492.

5)     Castagna, R., Di Donato, A., Strangi, G., Lucchetta, D.E. (2022) Light controlled bending of a holographic transmission phase grating. Smart Materials and Structures, 31(3), 03LT02.

6)     Vlasov, Y.A., Bo, X.-Z., Sturm, J.C., Norris, D.J. (2001) On-chip natural assembly of silicon photonic bandgap crystals. Nature, 414(6861), 289-293.

7)     Yu, Q.; Braswell, S.; Christin, B.; Xu, J.; Wallace, P. M.; Gong,H.; Kaminsky, D. Surface-enhanced Raman scattering on gold quasi-3D nanostructure and 2D nanohole arrays.Nanotechnology2010,21,355301−10.

Reviewer 2 Report

Reviewer report

NANOMATERIALS, Article

ID: nanomaterials-2549810

Title:  All-dielectric gratings with high-quality structural colors  

By Tongtong Wang, et al.

The manuscript submitted to Nanomaterials is devoted to study, by the computational method (using FDTD simulations) the HfO2 dual-layer grating over glass able to generate high-quality structural colors.  In simulations Authors considered relative geometrical factors of the gratings defined by 5 parameters: period P, thicknesses of the two components of the grating thickness H1 and H2 and their size D1 and D2 and surrounding refractive index aimed to determine quality, hue and chromaticity of structural colors generated upon white-light illumination.

Certainly, this interesting work deserves publication and is well suited to Nanomaterials journal.  The English language of the manuscript is correct with only few small deficiencies and the obtained results for HfO2 give promise that this dielectric metamaterial might be promising platform in novel devices and after commercialization also can dramatically improve the quality of our daily life by replacing various dyes and pigments.

Some general remarks regarding the manuscript:

1.      The whole work is basing on simulations, however, despite that no details of simulations are given except that Authors used FDTD method and exploit the Mie resonances. I would expect the details about programming platform that was used, mesh details like size, geometry (please explain what means mesh accuracy = 6 ?), equations used for calculation of the effective dielectric permittivity epsilon and magnetic permeability for the studied material. What type of boundary conditions has been assumed (PML on chosen surfaces of the grating or on all of them), and how the surface corrugation may influence the ideal-surface results presented in this work. The mentioned issues are necessary for the reproduction of the simulations by other researchers.

2.      There is no mention about the involved hafnia properties, like dispersion of the refractive index or indices and positions and width of the Mie resonances. The HfO2 adopts various structures including polar ones (piezoelectric and ferroelectric) and crystallize in various crystalline phases, moreover can be polycrystalline and can exhibit glassy phase. All these phases are described by different structural, optical and mechanical properties. Therefore, my question is what type of material was simulated? It is well known that dielectric and magnetic properties depend on the deposition method used the same is true for the surface corrugation which for hafnia is considerable and my influence the final results. Please comment in your work on the mentioned issues, and at least provide the reader with exact parameters used in calculations.

3.      I have difficulty to find out the coordinate axes geometry in Fig. 1, can you paste much larger 3D coordinate axes system? I do not understand Authors sentence on page 3,  “Due to the high symmetry of the structural unit in the X and Y directions, we imposed periodic boundary conditions in this direction.”  Do you impose periodic boundary conditions also in Y direction or only for the X direction? Along the Y direction there is no periodic structure. Please be more specific. Can you better define the properties of the used plane wave, is it linearly polarized beam with polarization along the grating wave vector or perpendicular to it? What is the spectral distribution of source light?

4.      Can you show in Fig. 5 also electric field energy distribution and the view along the z-axis and discuss the results in terms of light modes?

5.      Can you improve the quality of Fig. 6 by using much larger crosses signs, in black and white printing of your paper there are almost invisible? Can you give the FWHM whenever you chow the reflection peaks? In Fig. 7 in the figure caption, you describe FWHM and in the axis description it is HWHM is it the same or not? What you mean by the reflection rate?

6.      In Fig. 9 caption you show the reflectance spectra for different polarization angles. Please specify, whether you are using linear polarizer to read the reflectance spectra, and what means polarization 0 degree, with respect to what a grating wave vector or something else? Or maybe you are using polarized input beam, then describe the polarization of this beam. This experiment is not well described.

7.      In Conclusions Authors wrote (bottom of page 10) that “The physical mechanism of the reflection peak in the working range is elucidated using the magnetic dipole resonance mode”. I did not find in the work any place where this physical mechanism was elaborated.

Some minor remarks.

1.      I suggest to consider the change in all figure captions. Instead of writing “Figure1. illustrates the schematic diagram…” scientists use more frequently the form Figure 1. The schematic diagram….. Please read and improve all the figure caption in this fashion.

2.      On page 2, 5th line from the top remove bracket [.

3.      Authors, please consider change whenever you write “as show in Fig. x” into “as shown in Fig. x).

4.      In the last line of the Abstract, the sentence “This study holds immense significance…” could be written with less exaggeration.

Concluding my review, I am convinced that after improvements according to the proposed by me issues this paper can be accepted to the Nanomaterials journal, however at this stage I think that the minor/major revision is mandatory.

English language is correct, only minor corrections can be done.
